# Paediatric Deep Neck Infection—The Risk of Needing Intensive Care

**DOI:** 10.3390/children9070979

**Published:** 2022-06-29

**Authors:** Vojtech Perina, David Szaraz, Hana Harazim, Milan Urik, Eva Klabusayova

**Affiliations:** 1Department of Oral and Maxillofacial Surgery, University Hospital Brno, Faculty of Medicine, Masaryk University, Jihlavska 20, 625 00 Brno, Czech Republic; perina.vojtech@fnbrno.cz (V.P.); szaraz.david@fnbrno.cz (D.S.); 2Department of Paediatric Anaesthesiology and Intensive Care Medicine, University Hospital Brno, Faculty of Medicine, Masaryk University, Kamenice 5, 625 00 Brno, Czech Republic; harazim.hana@fnbrno.cz; 3Department of Simulation Medicine, Faculty of Medicine, Masaryk University, Kamenice 5, 625 00 Brno, Czech Republic; 4Department of Paediatric Otorhinolaryngology, University Hospital Brno, Faculty of Medicine, Masaryk University, Cernopolni 9, 662 63 Brno, Czech Republic; urik.milan@fnbrno.cz

**Keywords:** deep neck infection, paediatric, children, anatomy, complications, intensive care, airway management

## Abstract

Deep neck infections are potentially dangerous complications of upper respiratory tract or odontogenic infections. The pathophysiology, clinical presentation, and potential spreading depend on the complex anatomy of the neck fascia. These infections can lead to severe pathological conditions, such as mediastinitis, sepsis, and especially airway impairment with difficult management. Because of the risk of life-threatening emergency situations and the possible impacts on the overall health status of affected children, their early recognition is of utmost importance. Torticollis, drooling, and stridor are the most common signs of advancing disease. Children presenting with these symptoms should be admitted to the paediatric intensive care unit for vital function monitoring, where the airway could be readily secured if function is compromised.

## 1. Introduction

The head and neck region is relatively often affected by infectious diseases of various aetiologies and severity. They range from very common and benign conditions (superficial skin inflammations, localised odontogenic foci, uncomplicated tonsillitis) to more rare but potentially lethal conditions (abscesses and phlegmons of the deep cervical spaces) [1]. The incidence of deep neck infections (DNIs) is higher in the adult population than in children, and the predominant aetiology is also different [2,3]. In adults, odontogenic inflammation is the most common cause, whereas a dental cause is less common in children and complications of cervical lymphadenitis and upper respiratory tract infections predominate. This difference also reflects the common bacterial spectrum. In the paediatric population, infections caused by staphylococci and streptococci predominate, whereas, in adults, inflammation is mostly polymicrobial with a frequent anaerobic component [4,5,6,7]. Due to the more abundant presence of lymphoid tissue in infancy, children are at risk for the lymphogenic spread of infection. This is clinically significant, especially in retropharyngeal abscesses (lymph node colliquation is much more common than true abscesses but the line between these conditions is blurry. In clinical terminology, they are both referred to as abscesses. This article also uses this simplified terminology) that can spread caudally into the mediastinum. These children are at risk of serious complications such as mediastinitis, sepsis, and airway obstruction with the risk of difficult management. This review summarises the relevant anatomy of neck spaces, clinical symptoms, and basic treatment concepts of DNI. Warning signs and challenges in recognising and solving possible life-threatening complications are highlighted.

## 2. Relevant Anatomy

The clinical presentation, spread, and potential complications of cervical infections are determined by anatomical conditions. The principal determining factor is the arrangement of the fibroelastic cervical fascia, which has two basic layers. The superficial cervical fascia is relatively sparse, lies immediately below the subcutaneous tissue, envelops the platysma muscle, and, together with the attachments of the mimic muscles, forms the superficial musculoaponeurotic system. Infections below the superficial layer of the cervical fascia are not considered to be deep cervical infections [8].

### 2.1. Deep Cervical Fascia

The deep cervical fascia (DCF) envelops the neck muscles, salivary glands, blood vessels, nerves, lymphatic tissue, and other neck organs (thyroid, larynx, pharynx, oesophagus). It provides mechanical support and allows the reciprocal movement necessary for head movement and the act of swallowing. The DCF consists of three layers; superficial (investing), pretracheal, and prevertebral.

The superficial layer of the DCF surrounds the entire neck, dorsally attached to the spinous process of C7 vertebra and ligamentum nuchae; caudally to the scapula, clavicle, and manubrium sterni; cranially to the inferior edge of the mandible, zygomatic arch, mastoid process, and occipital bone; and anteriorly to the hyoid bone. After division, it envelops the trapezoid muscle, sternocleidomastoid muscle, the anterior belly of the digastricus muscle, the submandibular salivary gland, the masseter muscle, and the parotid gland. Below the sternocleidomastoid muscle, the lateral part of the carotid sheath and the cranial part of the superficial layer of the DCF covering the anterior belly of the digastric and mylohyoid muscles form the floor of the submandibular space. During inflammation, the superficial layer of the DCF prevents the passage of pus into the subcutaneous tissue and the formation of an external fistula; therefore, an abscess could instead spread into deeper cervical spaces.

The deep layer of the DCF covers the spine and deep neck muscles. Dorsally, it attaches to the spinous processes of the cervical vertebrae. On the ventral side of the cervical spine (near the longus colli muscle), it divides into a prevertebral leaf, firmly adhering to the anterior side of the vertebrae (from the base of the skull to the level of Th3) and a looser alar leaf which anteriorly lies on the buccopharyngeal fascia and allows the movement of the pharynx during swallowing.

The middle layer of the DCF consists of two parts, muscular and visceral. The muscular part surrounds the infrahyoid muscles, attaches to the hyoid cranially and the sternum caudally, and is relatively firm. The visceral part encloses the thyroid gland, oesophagus, pharynx, larynx, and trachea. Caudally, it extends into the mediastinum and covers the pericardium and aortic arch. Cranially, it attaches anteriorly to the hyoid bone; dorsally, it forms the buccopharyngeal fascia and part of the carotid sheath. The carotid sheath is formed by the union of the superficial and pretracheal fascia ventrally and the prevertebral fascia dorsally. It contains the common and internal carotid artery, internal jugular vein, and the vagal nerve [9,10].

The anatomical nomenclature of the cervical fascia and the spaces defined by it is not completely uniform and differs in detail between different publications. The anatomical spaces described below are also not free spaces in the true sense of the word, but potential spaces that only become clinically important in pathological conditions [11,12,13,14].

### 2.2. Deep Neck Spaces and Their Infections

The anatomical arrangement and attachments of the individual layers of the cervical fascia create preformed spaces. During infectious diseases, pus is initially confined to one space, but as the disease progresses, the natural barrier, if any is present, between the spaces is overcome and the inflammation spreads. The clinically significant barrier is the hyoid bone and its attachments to the superficial and visceral layer of the DCF. The deep cervical spaces can be divided into suprahyoid, infrahyoid, and spaces occupying the entire length of the neck according to their relationship to the hyoid.

The only clinically relevant infrahyoid space is the pretracheal/anterior visceral space, which is enclosed by the visceral portion of the middle leaf of the DCF. Infections of this space arise by spreading from the submandibular space or trauma of the oesophagus. They are often associated with emphysema of the neck after perforation of the oesophagus. Oedema of the larynx and the compromising of airways are common. The infection of this space spreads easily to the mediastinum, and the patient’s condition may deteriorate rapidly without previous warning signs [8,15].

The suprahyoid spaces represent the anatomically and clinically most complex group of cervical spaces. Abscesses of the facial spaces around the ramus mandibulae (buccal, parotid, and masticatory space) are dangerous because of the possibility of the spread of the infection to the parapharyngeal space and because of frequent severe trismus that complicates airway management, but they do not cause airway obstruction per se. The source of infections in these spaces is odontogenic or sialadenitis.

Infections of the spaces bordered by the body of the mandible (submental, sublingual, and submandibular space) have the same source. The sublingual and submandibular spaces communicate freely with each other beyond the dorsal margin of the mylohyoid muscle and open into the parapharyngeal space. Infections of these spaces have a substantial risk of spreading parapharyngeally, and a larger abscess may result in the craniodorsal dislocation of the tongue and may compromise airway patency. This is especially prominent for the severe form of bilateral cellulitis of the submandibular spaces—Ludwig’s angina. It is manifested by the brawny and tender swelling of the floor of the mouth and neck with the rapid development of respiratory distress and a septic state. It is especially dangerous for immunocompromised patients [16,17,18,19]. Related to Ludwig’s angina is cervical necrotising fasciitis. It is characterised by the rapid spreading of infections in the subcutaneous tissue and superficial cervical fascia. Although it is rare in children (an uncommon disease leading to tissue destruction and necrotising fasciitis occurring only in the paediatric population is noma, a polymicrobial devastating infection of the head and neck region occurring in malnourished, immunocompromised children in sub-Saharan Africa [20]) it is dangerous because of mild initial symptoms with a later precipitous deterioration of clinical state. Septic shock and mediastinitis are common among patients suffering from necrotising fasciitis [21,22].

Abscesses of the peritonsillar space complicate tonsillitis and are at risk of spreading to the para- and retro-pharyngeal space. There is controversy in their treatment as to when to perform tonsillectomy in the acute stage, when to postpone it, and when to not perform it at all [23]. The lateral pharyngeal space is, clinically, a particularly important space. It has the shape of an inverted square-based pyramid, with its base at the skull base and apex at the greater corn of the hyoid bone. It communicates ventrally with the submandibular space and dorsomedially with the retropharyngeal space. Sources of infection of the parapharyngeal space are diverse and include pharyngitis, tonsillitis, otitis, mastoiditis, parotitis, or cervical lymphadenitis, as well as odontogenic sources when spread from surrounding spaces. The clinical picture differs according to the localisation of infection related to the styloid septum. When the prestyloid part of the parapharyngeal space is involved, then trismus and sore throat are common. When the retrostyloid part is affected, clinical problems are less significant, but serious complications may occur when the infection spreads to the contents of the carotid sheath. These include infective internal jugular vein thrombosis (Lemierre’s syndrome), carotid artery aneurysm, Horner’s syndrome, cranial nerve IX–XII palsies, and mediastinitis and septicaemia [24,25,26,27].

The aforementioned carotid sheath runs throughout the entire length of the neck, in which infections of the prevertebral space are uncommon and have little tendency to spread. Furthermore, between the prevertebral fascia and the dorsal wall of the pharynx and oesophagus, the retropharyngeal space is located ventrally and the so-called danger space dorsally. They are separated by the alar fascia [28]. The retropharyngeal space ends caudally at the level of the Th3 vertebra and is divided into left and right halves by the adhesion of the alar and buccopharyngeal fascia in the midline. Its infections of rhinogenic origin are common in young children (up to 4–5 years of age) because it contains lymph nodes that disappear at around 6 years of age [29,30]. In older patients, this space may become infected by transfer from the parapharyngeal space or after injury to the posterior wall of the pharynx. Patients with retropharyngeal abscess are at risk of airway obstruction and the aspiration of pus after abscess rupture. The danger space is distal to the retropharyngeal space, extends cranially to the skull base, caudally down to the diaphragm, and contains loose areolar tissue, and therefore, its infections have a tendency to spread rapidly to the mediastinum. The danger space is infected secondarily by the spread of infection from the parapharyngeal, retropharyngeal, or prevertebral space. The phenomenon of false recovery is a major concern, where the patient’s discomfort subsides after the pressure in the retropharyngeal or parapharyngeal space is relieved by the evacuation of pus into the danger space. After a short interval, however, sepsis, mediastinitis or even mediastinal empyema may develop [31]. There is a summary of the anatomy, spreading, and risks of particular deep neck spaces in Table 1 and in Appendix A.

## 3. Age-Related Specificities

In the whole population under 18 years of age, the most common antecedent illnesses are upper respiratory tract infection, followed by dental infection and congenital anomalies [32]. In the younger paediatric population, the most common DNIs are of respiratory, and subsequent lymphatic, origin, mainly affecting the retropharyngeal space. Retropharyngeal abscesses tend to be more common in children about 4 years of age due to the frequency of upper respiratory infections and the presence of retropharyngeal lymph nodes. Peritonsillar and parapharyngeal abscesses are more frequent in children older than 4 years and peritonsillar abscesses are the most frequent in adolescents or young adults. The lymphatic origin of deep neck abscesses is more common among younger children because they frequently suffer from lymphadenitis. On the other hand, odontogenic infections are related to dental caries or wisdom teeth eruption and are thus common in older children and adolescents. There is a higher incidence of DNIs among boys [2,24,33,34,35,36,37,38,39,40,41]. See Table 1 for specific information about age-related predominance in abscess localisation. 

## 4. Basic Microbial Findings

The microbiological profile of DNIs changes with the age of patients. Younger children are predominantly infected by respiratory pathogens, Streptococcus pyogenes and β-lactamase secreting Staphylococcus aureus being the most common. With increased age, Gram-negative and anaerobic flora are more common (odontogenic infections) [4,6,7,33,37,42,43]. A rising incidence of methicillin-resistant Staphylococcus aureus (MRSA) has been reported. Its spread and prevalence vary among different countries and populations. However, patients infected with resistant strains have a greater risk of severe disease course, including mediastinal involvement [42,44,45,46,47,48].

Less frequent are specific cervical inflammations as a consequence of lymph node colliquation caused by the relatively frequent, in children (20%), extrapulmonary manifestation of tuberculosis, non-tuberculous mycobacteria, or actinomycetes [49,50,51,52] (a serious incident is the occurrence of a local epidemic of nosocomial Mycobacterium abscessus infection in children after dental treatment. The cause was contaminated dental unite cooling water [53,54]).

## 5. Diagnostics and Management of Uncomplicated DNIs

The most common symptoms of DNIs at presentation are generally fever, torticollis, neck swelling or mass, sore throat, odynophagia, and decreased oral intake. Children less than 2 years of age present with fewer and less specific symptoms, such as fever, lethargy, or irritability, so the diagnosis may be difficult [55,56]. Children younger than 4 years of age are more likely to present with agitation, drooling, rhinorrhoea, stridor, and even respiratory distress. Trismus, “hot potato voice”, and other localising signs are seen more often in older children [9,23,31,32,57]. The initial diagnosis may be delayed in younger or uncooperative children because of limited verbal communication, difficulties in completing a comprehensive head and neck physical examination, and the similarity of clinical symptoms with common respiratory infections [58].

Some clinical findings are specific to the particular space involved. Facial swelling and trismus appear with the involvement of masticatory, buccal, parotid, submandibular, and sublingual spaces. Trismus and torticollis are common in parapharyngeal abscesses. The displacement of the tonsils and uvula is typical for peritonsillar abscesses and the bulking of the pharyngeal wall medially occurs in danger space and paramedially in retropharyngeal space abscesses [8,57,59,60].

Diagnostic imaging is used for the precise localisation of inflammation and mainly for differentiating cellulitis from abscesses. Ultrasound as a simple imaging technique may be sufficient in cooperative patients with superficially and laterally localised lesions. In those situations, ultrasound has a similar sensitivity to CT [61]. MRI could be considered because of the absence of ionising radiation, but it requires general anaesthesia (GA) more frequently than CT or ultrasound. The accuracy of CT and MRI is comparable in diagnosing deep neck abscesses. CT or MRI should be preferred in patients with an aggravating clinical course or compromised airways [61,62,63,64,65,66]. 

In an uncomplicated course of neck infection, there are no specific laboratory findings, different to other bacterial infectious diseases. White blood cell count and C-reactive protein levels are elevated. A neutrophil-to-lymphocyte ratio higher than five seems to be an indicator of advanced inflammation [67,68,69,70].

In the absence of airway compromise or severe clinical status deterioration, therapy should start with an empiric IV. antibiotic course for 48 h; Amoxicillin/Clavulanic acid 50 mg/kg eight-hourly is preferred. It is curative in cellulitis and abscesses smaller than 2.5 cm in diameter. Conservative treatment is effective in cellulitis where the vascular supply is preserved and antibiotics are more effective [71]. However, children under 4 years of age require surgical drainage more often. Moreover, CT/MRI for proper diagnosis usually requires GA in this age group. Therefore, it is recommended to perform needle aspiration or surgical drainage of the suspected mass in the same GA as diagnostic imaging. When 48 h IV antibiotic therapy does not improve the clinical status, in patients with abscesses larger than 2.5 cm and younger than 2 years, surgical drainage is recommended. Immediate surgery is necessary when dyspnoea or another complication occurs [3,18,24,30,37,39,41,58,67,71,72,73,74,75,76,77]. 

## 6. Possible Complications and Their Warn Sings

The rate of complications of paediatric DNIs is 5–10%; airway compromise, multiple neck space involvement, and mediastinitis are the most common. Rare but severe events are pus aspiration, internal jugular vein thrombosis, internal carotid pseudoaneurysm, Ludwig’s angina, and sepsis [3,23,24,37,45,72,78,79,80,81]. Generally, children younger than 2 years with retropharyngeal or multiple space abscesses are the most endangered by complications [45,67,79,82]. The high risk of complications, impending airway compromise, and its difficult management must be anticipated in those children.

### 6.1. Airway Obstruction

The complication of the most concern is acute airway obstruction. Warning signs that necessitate early airway intervention include marked tachypnoea with shallow respiration, the use of accessory respiratory muscles, orthopnoea, dyspnoea, stridor, and the adoption of a sniffing position [19]. However, the majority of cases can be successfully managed with endotracheal intubation. The decision to secure the airway by tracheal intubation should not be delayed because of the possibility of difficult airway management if the condition worsens (swelling, oedema) and the unacceptable risks and complications related to acute tracheostomies in children [83,84,85].

Difficult airway management should be always anticipated in DNIs. Moreover, there is a risk of airway collapse during anaesthesia induction. Therefore, all possible preventive precautions are required before airway management and/or anaesthesia induction. The need for IV access before initiating airway management and the preparedness of all necessary equipment (videolaryngoscope, flexible bronchoscope) is of utmost importance. Although there are no guidelines on the choice of anaesthesia induction technique, maintaining spontaneous ventilation during the period of airway securing is usually a preferred technique. This can be achieved by both inhalation and intravenous induction (preferably with anaesthetics that do not affect respiratory drive such as ketamine or the careful titration of remifentanil). The intubation technique of choice is primarily videolaryngoscopy or awake flexible intubation, and in all cases, the presence of an ENT surgeon capable of performing a tracheostomy on-site is recommended [86,87]. The meticulous monitoring of the airway should continue at the PICU for at least 48 h after surgical intervention because of the potential risk of increasing oedema in the postoperative period [8,18,19,88,89].

### 6.2. Vascular Complications

Vascular complications, such as carotid artery pseudoaneurysm, should be considered in children with severe headache, bruits, lower cranial nerve palsies, Horner’s syndrome, and nasal or oropharyngeal bleeding [26,81,90,91]. Internal jugular vein thrombosis with the embolisation of infected thrombi, Lemierre’s syndrome, or postanginal sepsis is a rare but dangerous condition. It is characterised by the septic thrombophlebitis of the internal jugular vein and at least one focus of septic embolus. Patients usually have a recent history of an oropharyngeal infection caused by an anaerobic pathogen, Fusobacterium necrophorum. Septic embolisation could cause systemic complications depending on the final end-point of the embolus. Pulmonary impairment is the most common and could lead to respiratory failure. Joint, liver and brain emboli with neurological complications or sepsis/septic shock are also possible.

### 6.3. Rapidly Spreading Inflamations

In the presence of tight, shiny, painful skin without circumscribed swelling that spreads rapidly in the submandibular space and on the front of the neck, the development of Ludwig’s angina or necrotising fasciitis is suspected. These dangerous disease states have their origin in the phlegmonous inflammation of the superficial layers of the tissues of the floor of the mouth and throat. Later, bullae and skin necrosis occur. After a period of mild symptoms, these patients are at risk of the rapid development of toxic shock, airway obstruction and the spread of infection to the mediastinum. Radical surgical drainage, securing the airway, and intensive care are immediately indicated [22,92,93]. 

### 6.4. Sepsis

Continuous fever ≥39.0 °C, CRP ≥ 50 mg/L, WB cell count ≥15.0 × 10^9^/L, procalcitonin >2 ng/mL, dysphagia, odynophagia, dysphonia, neck pain, and limited neck movement also indicate inflammation progression. Torticollis, excessive drooling, stridor, and respiratory distress are signs that require particular attention. These children should be strictly monitored and early admission to the paediatric intensive care unit (PICU) is recommended. This is vital in children presenting with signs of sepsis and/or systemic inflammatory response syndrome. Prolonged capillary refill time (>3 s), mottled and cool extremities, altered mental status, oliguria, and/or hypotension are the most common [5,38,44,64,75,76,80,82,94,95,96,97,98].

## 7. Differential Diagnosis of Neck Masses

Neck masses in children usually belong to one of three categories: developmental, inflammatory/reactive, or among tumours. Non-severe reactive lymphadenopathy is common in children, the reason being frequent antigenic stimulation. Neck lymphadenopathy is also one of the symptoms of Kawasaki disease and Paediatric Inflammatory Multisystem Syndrome associated with COVID-19. Neck abscesses can also result from colliquated infectious lymphadenitis (bacterial and mycobacterial infections, cat-scratch disease). The most common congenital neck lesions include thyroglossal duct cysts, branchial arches cysts, dermoid cysts, vascular malformations, and haemangiomas. Common benign tumours include lipomas, fibromas, and neurofibromas. Malignant lesions are rare in children. Malignancy is suspected when the mass is hard or rubbery in consistency, fixed to the surrounding area, and greater than 2 cm in diameter. Moreover, a mass persisting for more than two weeks and unresponsive to antibiotic treatment is suspicious of a malignant process. Abscesses and infectious lymphadenopathy are accompanied by clinical and laboratory signs of inflammation, which distinguishes them from congenital malformations and tumours. However, persistent fever may accompany lymphomas or leukaemias as one of the B symptoms (other B symptoms include night sweats and loss >10% of body weight in a period of 6 months) [22,99,100,101,102,103,104].

## 8. Conclusions

Upper respiratory tract infections and subsequent lymphadenopathies are very common among children. Most are of viral aetiology, are mild and self-limiting, and require no inpatient treatment; these facts could lead to the underestimation of the severe complications that may occur [105]. Progression to an abscess in deep neck spaces is uncommon but could result in life-threatening complications in more than 10% of patients [106]. The most dangerous are descending mediastinitis and airway obstruction. The early recognition of imminent danger, prompt airway management, and appropriate intensive care are crucial for children endangered by DNIs [107,108] Admission to the PICU is recommended, especially for children with signs of impending respiratory damage/respiration sepsis or conditions that may progress rapidly. These are, particularly, danger space abscesses, carotid sheath involvement, and phlegmonous inflammation. In the intensive care unit, thanks to meticulous monitoring of the patient, the deterioration of the clinical condition can be detected without delay. Specialists trained in securing the airway in difficult conditions and the relevant equipment are also usually more readily available than in standard wards [82,109].

## Figures and Tables

**Table 1 children-09-00979-t001:** Deep neck spaces.

		Boundaries						Content	Source	Spreading to	Risks	Predominant Age
		Superior	Inferior	Dorsal	Ventral	Medial	Lateral					
Infrahyoid	Pretracheal/anterior visceral	Thyroid cartilage	Direct communication with superior mediastinum	Esophagus, anterior wall	Middle layer of DCF, visceral division	Middle layer of DCF, visceral division	Middle layer of DCF, visceral division	Thyroid gland, trachea	Esophagus perforation	Superior mediastinum	High risk of airway compromise, mediastinitis	Any
Suprahyoid	Peritonsillar	Anterior and posterior tonsillar pillars connection	Not defined	Posterior tonsillar pillar	Anterior tonsillar pillar	Tonsil	Superior pharyngeal constrictor	Loose connective tissue	Tonsilitis	Parapharyngeal, retropharyngeal space	Moderate risk of airway compromise. Trismus	Older children (12y)
	Sublingual	Mucosa of oral floor	Mylohyoid m.	Parapharyngeal space	Mandible	genioglossal, geniohyoid m.	Mandible	Sublingual gland, lingual and hypoglossal nerves	Sialadenitis, odontogenic (decidual teeth, M1 and mesial teeth)	Parapharyngeal, submandibular space	Moderate risk of airway compromise. Tongue dislocation	Related to teeth eruption and caries
	Submandibular	Mandible, mylohyoid m.	Superficial layer of DCF	Parapharyngeal space	Anterior belly of digastricus	Anterior belly of digastricus	Mandible	Submandibular gland, lymph nodes	Odontogenic-M2, M3	Parapharyngeal, sublingual space	Trismus, spreading	Older children—related to teeth eruption and caries
	Submental	Mylohyoid m.	Superficial layer of DCF	Hyoid bone	Superficial layer of DCF	Not defined	Anterior belly of digastricus	Lymph nodes	Frontal teeth	Submandibular space	Spreading	Rare
	Parapharyngeal	Skull base	Hyoid bone	Prevertebral fascia, carotid sheath	Pterygomandibular raphe	Middle layer of DCF, visceral division	Superficial layer of DCF, medial pterygoid m.	Styloid septum, maxillary artery and nerve, adipose tissue	Tonsilitis, odontogenic (M3), lymph nodes, sialadenitis, mastoid abscess, other spaces	Retropharyngeal space, Danger space, carotid sheath	Trismus, frequent spreading	Younger children (6y)—poststyloid part
Length of whole neck	Carotid sheath	Skull base	Mediastinum	Prevertebral fascia	Superficial layer of DCF	Middle layer of DCF, visceral division	Superficial layer of DCF	Carotid artery, internal jugular vein, cervical sympathetic chain, cranial nerves IX, X, XI, XII.	Parapharyngeal space, intravenous	Mediastinum, intracranially	Internal jugular vein thrombosis, carotid aneurysm, Horner’s palsy	Rare, younger children (6y)
	Retropharyngeal	Skull base	Mediastinum, Th2 level	Alar fascia	Buccopharyngeal fascia	Midline fusion	Carotid sheath	Lymph nodes	Lymph nodes, trauma, parapharyngeal space	Carotid sheath, parapharyngeal space	High risk of airway compromise. Pus aspiration	Younger children (4–5y)
	Danger	Skull base	Mediastinum, diaphragm level	Prevertebral fascia	Alar fascia	Not defined	Fused fascias on cervical vertebrae	Loose areolar tissue	Retropharyngeal, parapharyngeal, prevertebral space	Mediastinum	Mediastinitis, sepsis. False recovery	Any
	Prevertebral	Skull base	Coccyx	Vertebral bodies, deep muscles	Prevertebral fascia	Not defined	Prevertebral fascia fusion to cervical vertebrae	Dense fibrous tissue	Retropharyngealand danger space, pharynx perforation	Spreading limited due to stiff tissue	Spinal osteomyelitis and instability	Any
Spaces of the face	Masticatory muscles space (masseter, medial pterygoid and temporal)	Skull base	Submandibular space, mandible	Parotid space, parapharyngeal space	Buccal space	Medial pterygoid m. + superficial layer of DCF	Masseter m. + superficial layer of DCF	Temporalis m., mandibular nerve, internal maxillary artery	Odontogenic-M3	Buccal, parotid, submandibular, parapharyngeal space. Orbit.	Trismus	Adolescents
	Buccal	Zygoma	Mandible	Pterygomandibular raphe	Not defined	Buccopharyngeal facia	Skin of the cheek	Buccal fat pad, parotid duct, facial artery	Odontogenic	Masticatory muscles space	Facial cellulitis	Any
	Parotid	Parotic capsula *	Parotic capsula *	Parotic capsula *	Parotic capsula *	Parotic capsula *	Parotic capsula *	Parotid gland, facial nerve, external carotid artery, lymph nodes	Sialadenitis, odontogenic	Parapharyngeal space	No severe	Any

* Parotic capsula is derivative of the superficial layer of DCF.

## Data Availability

Not applicable.

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
