# Peer review of "Paediatric Deep Neck Infection—The Risk of Needing Intensive Care"

_children, 2022, doi:10.3390/children9070979_

Round 1

Reviewer 1 Report

A good summary of the anatomy of the deep spaces of the neck. The paper however, does not really add anything new to the literature. In considering possible complications of deep space neck infections, the authors failed to include the possibility of vascular compression/clots of the internal jugular vein (Lemierre's syndrome) with possible septic emboli consequences. 

There are some spelling/grammar/wording errors (examples outlined below) 

- page 2 line 61 the superficial layer of the DCF..... instead of "distally" this should read "dorsally"

- usually in English it would be trapezius muscle etc (not m. trapezius)

Author Response

A good summary of the anatomy of the deep spaces of the neck. The paper however, does not really add anything new to the literature.

          The aim of our article is to draw the attention to the specific problem of deep cervical inflammation to specialists (paediatricians, paediatric intensivists) who do not commonly encounter this problem. Therefore, much space is devoted to the relevant anatomy, which is complex and influences the manifestation and clinical course of the disease. The article is conceived as a summarising review; the ENT specialist or maxillofacial surgeon will not find much new information in it, but these specialists are not the target group for whom the article is intended.

In considering possible complications of deep space neck infections, the authors failed to include the possibility of vascular compression/clots of the internal jugular vein (Lemierre's syndrome) with possible septic emboli consequences.

          We agree with this comment. Lemierre's syndrome was mentioned only in the section on the anatomy of the carotid sheath (lines 141-143). Which was not sufficient. Information on this disease has been added to section 6 (lines 276 – 283).

There are some spelling/grammar/wording errors (examples outlined below)

- page 2 line 61 the superficial layer of the DCF..... instead of "distally" this should read "dorsally"

- usually in English it would be trapezius muscle etc (not m. trapezius)

          These errors have been fixed

Reviewer 2 Report

This is an interesting review about pediatric deep neck infection and the risk of needing intensive care.

The paper is well written. However, some issues remain.

A summarizing table should be included in the paper.

The authors should add some data about differential diagnosis of cervical masses in children and about causes of colliquated adenopathies, such as atypical micobacteriosis.

Author Response

This is an interesting review about pediatric deep neck infection and the risk of needing intensive care. The paper is well written. However, some issues remain.

A summarizing table should be included in the paper.

          The table summarizing deep neck spaces boundaries, infection spreading, risks and age predominance of their abscesses is attached: Table 1 (file: table_spaces.xlsx)

The authors should add some data about differential diagnosis of cervical masses in children and about causes of colliquated adenopathies, such as atypical micobacteriosis.

          This is a very relevant comment. Information’s were added – lines 191 – 193. And new section “Differential diagnosis of neck masses” was created.

Reviewer 3 Report

The manuscript of the "Paediatric deep neck infection – the risk of needing intensive care" is an informative review of deep neck infection. I would recommend publishing this manuscript after revising it.

However, because this topic is significant in educating and knowledge updating, I would ask the author to revise it to make it a great review. Here is a few suggestions.

1. I strongly recommend authors use a neck anatomy diagram to explain the anatomy of the arrangement of the fibroelastic cervical fascia, the position related to subcutaneous tissue and neck organs, and how they attach to muscles, especially where the DNI happen and spread. Ideally, if authors can show the anatomy of pediatric deep neck space that specifically facilitate certain types of infection spreading, that would be great. 

2. As this manuscript is titled "Paediatric", I would expect more content on pediatric infection. so section 3 (line 164- 172) content needs to expand to more details to emphasize the specificities in the pediatric population. and explore possible reasons for each age group and gender preference. 

3. I also would like to see more content on why the pediatric DNI needs Intensive care. 

Author Response

The manuscript of the "Paediatric deep neck infection – the risk of needing intensive care" is an informative review of deep neck infection. I would recommend publishing this manuscript after revising it. However, because this topic is significant in educating and knowledge updating, I would ask the author to revise it to make it a great review. Here is a few suggestions.

I strongly recommend authors use a neck anatomy diagram to explain the anatomy of the arrangement of the fibroelastic cervical fascia, the position related to subcutaneous tissue and neck organs, and how they attach to muscles, especially where the DNI happen and spread. Ideally, if authors can show the anatomy of pediatric deep neck space that specifically facilitate certain types of infection spreading, that would be great.

          We agree that a diagram would be helpful. To facilitate review process, we are now submitting revised version of text and meanwhile second round of reviews we try to make such diagram.

As this manuscript is titled "Paediatric", I would expect more content on pediatric infection. so section 3 (line 164- 172) content needs to expand to more details to emphasize the specificities in the pediatric population. and explore possible reasons for each age group and gender preference.

          The section 3 was expanded. Relevant information can be also find in the table 1

I also would like to see more content on why the pediatric DNI needs Intensive care.

          Some comments were added into conclusion section

Round 2

Reviewer 3 Report

The revised version provides much more information on DNI and focused on the pediatric population. Thank you for the timely work.